# Ballistic edge states in Bismuth nanowires revealed by SQUID interferometry

Anil Murani[1], Alik Kasumov[1,2], Shamashis Sengupta[1], Yu A. Kasumov[2], V.T. Volkov[2], I.I. Khodos[2], F. Brisset[3], Raphaëlle Delagrange[1], Alexei Chepelianskii[1], Richard Deblock[1], Hélène Bouchiat[1] & Sophie Guéron[1]

The protection against backscattering provided by topology is a striking property. In two-dimensional insulators, a consequence of this topological protection is the ballistic nature of the one-dimensional helical edge states. One demonstration of ballisticity is the quantized Hall conductance. Here we provide another demonstration of ballistic transport, in the way the edge states carry a supercurrent. The system we have investigated is a micrometre-long monocrystalline bismuth nanowire with topological surfaces, that we connect to two superconducting electrodes. We have measured the relation between the Josephson current flowing through the nanowire and the superconducting phase difference at its ends, the current–phase relation. The sharp sawtooth-shaped phase-modulated current–phase relation we find demonstrates that transport occurs selectively along two ballistic edges of the nanowire. In addition, we show that a magnetic field induces $0-\pi$ transitions and $\varphi_0$-junction behaviour, providing a way to manipulate the phase of the supercurrent-carrying edge states and generate spin supercurrents.

[1] Laboratoire de Physique des Solides, CNRS, Univ. Paris-Sud, Université Paris-Saclay, 91405 Orsay Cedex, France. [2] Institute of Microelectronics Technology and High Purity Materials, RAS, 6, Academician Ossipyan str., Chernogolovka, Moscow Region 142432, Russia. [3] Institut de Chimie Moléculaire et des Matériaux d'Orsay, Univ. Paris-Sud, Université Paris-Saclay, CNRS UMR 8182, 91405 Orsay Cedex, France. Correspondence and requests for materials should be addressed to H.B. (email: helene.bouchiat@u-psud.fr) or to S.G. (email: sophie.gueron@u-psud.fr).

Reducing the size of a conductor usually decreases its conductivity because of the enhanced effect of disorder in low dimensions, leading to diffusive transport and to weak, or even strong localization. Notable exceptions occur when topology provides protection against disorder, such as in the quantum Hall effect or the recently discovered quantum spin Hall effect in two-dimensional (2D) topological insulators[1,2]. In the latter, crystalline symmetry combined with high spin–orbit coupling generate band inversion and one-dimensional (1D) chiral edge states with perfect spin-momentum locking, that theoretically precludes backscattering along the edges. However, since the first evidences of edge state currents, both in the normal and superconducting-proximitized states, demonstrating the robustness of ballistic conduction and spin polarization in the 1D edge states has remained a challenge[3–7]. Here we provide a direct signature of ballistic 1D transport along the topological surfaces of a monocrystalline bismuth nanowire connected to superconducting electrodes. To this end, we measure the relation between the Josephson current $I_J$ flowing through a nanostructure and the superconducting phase difference $\varphi$ at its ends, the current–phase relation (CPR)[8]. The CPR is an exquisite tool to discriminate between different transport regimes (ballistic, diffusive, tunnel).

The sharp sawtooth-shaped CPR we find demonstrates that transport of Cooper pairs occurs ballistically[9–13] along two edges of the nanowire, whose positions can be deduced from experiments in different magnetic field orientations. We also show that a Zeeman field induces $0–\pi$ transitions[14] and $\varphi_0$-junction behaviour[15], in agreement with recent theoretical predictions[16].

## Results

**Investigated samples**. Monocrystalline bismuth nanowires were grown by sputtering (Methods section), and their crystalline orientation was determined by electron backscatter diffraction. Individual nanowires with (111) top facets were then selected to exploit the predicted topological edge states of those surfaces[17], brought about by the Bi lattice symmetries and high (eV-range) atomic spin–orbit coupling. The wires are narrow enough (thickness and width between 30 and 200 nm) that most normal state conduction is due to surfaces and edges, with only a few per cent contribution of the bulk states[18,19], as discussed in Supplementary Note 2. Furthermore, tight-binding simulations of a Bi nanowire with a rhombic section and top and bottom (111) facets, similar to the main nanowire reported here (Fig. 1 and Methods section), predict one edge state along each (111) facet, extending the result of Murakami[17]. These edge states coexist with 2D metallic states on the non-topological (100) surfaces but nonetheless clearly dominate the local density of states (Fig. 1c).

**Ballistic CPR**. To measure the CPR of the superconductor/bismuth nanowire/superconductor (S/Bi/S) junction, we inserted a single-Bi nanowire with (111) surfaces into an asymmetric SQUID set-up. The set-up consists of a high-critical-current Josephson junction, made of a $W$ superconducting nanoconstriction, in parallel with a 1-μm-long S/Bi/S junction characterized beforehand (Figs 1 and 3). In this configuration the modulation of the SQUID's critical current by the magnetic flux yields the CPR of the junction with the smallest critical current[20]. It is well known that the CPR of the superconductor/insulator/superconductor (SIS) Josephson junction is sinusoidal $I_J(\varphi) = I_c \sin\varphi$. The CPR of a ballistic superconductor/normal metal/superconductor junction, on the other hand, is a characteristic sawtooth in a long junction (for which $L \gg \hbar v_F/\Delta$)[9–13], or segments of a sine in a short junction

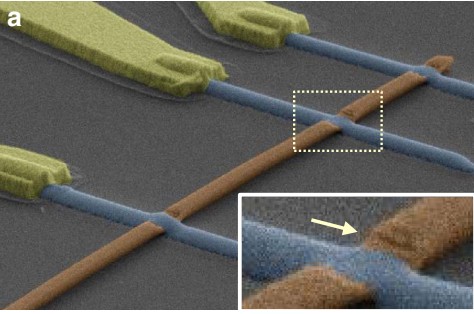

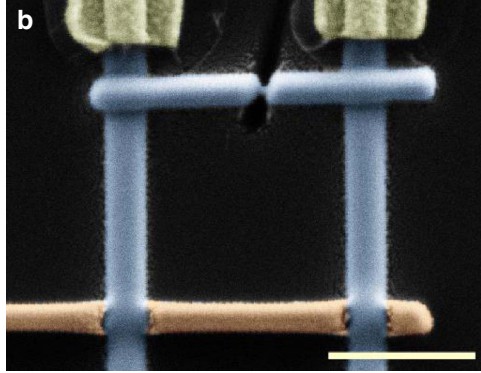

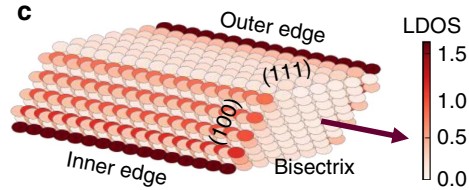

**Figure 1 | Building an asymmetric SQUID around a (111) Bi nanowire.** (**a**) Scanning electron micrograph of the bismuth nanowire with (111) top and bottom facets (brown), connected to high-critical-field superconducting tungsten electrodes (blue). The segment closest to the wire end forms the S/Bi/S junction that will be integrated in the asymmetric SQUID configuration. The supercurrents induced at low temperature through this S/Bi/S junction and others are described in Supplementary Table 1, Supplementary Fig. 6 and Supplementary Note 2. Inset: zoom of the contact area, explaining the asymmetry between the two conduction channels. The (111) surface has been etched by the FIB during tungsten deposition whereas the bottom surface is probably not damaged and thus better coupled to the tungsten superconducting wire. (**b**) Micrograph of the asymmetric SQUID built after characterizing the S/Bi/S junction: a high-critical-current tungsten nanoconstriction is in parallel with the smaller-critical-current S/Bi/S junction of **a**. Scale bar, 1 μm. (**c**) Tight-binding calculation of the local density of states (LDOS) of an infinitely long, five bilayer-thick (111) bismuth nanowire with rhombic section.

(Fig. 2c)[20] ($L$ is the length of the normal metal, $v_F$ the Fermi velocity and $\Delta$ the superconducting gap). Any disorder smooths the CPRs, as illustrated in Fig. 2c for a junction in the diffusive regime, (that is, with a mean free path shorter than the length $L$). While the CPR of atomic point contacts[20], nanowire-based quantum dots[21–23] and graphene[24] have previously been measured in this way, the CPR of micrometre-long, quasi-ballistic channels has, to our knowledge, not been accessed.

The switching current of the asymmetric Bi-SQUID (Fig. 2a) clearly displays sawtooth-shaped oscillations of amplitude 400 nA, superimposed on the 80 μA critical current of the nanoconstriction. The oscillation period of 9.5 G corresponds to a flux quantum $\Phi_0 = h/2e$ through the SQUID loop area of 2 μm². Those oscillations demonstrate the sawtooth CPR characteristic

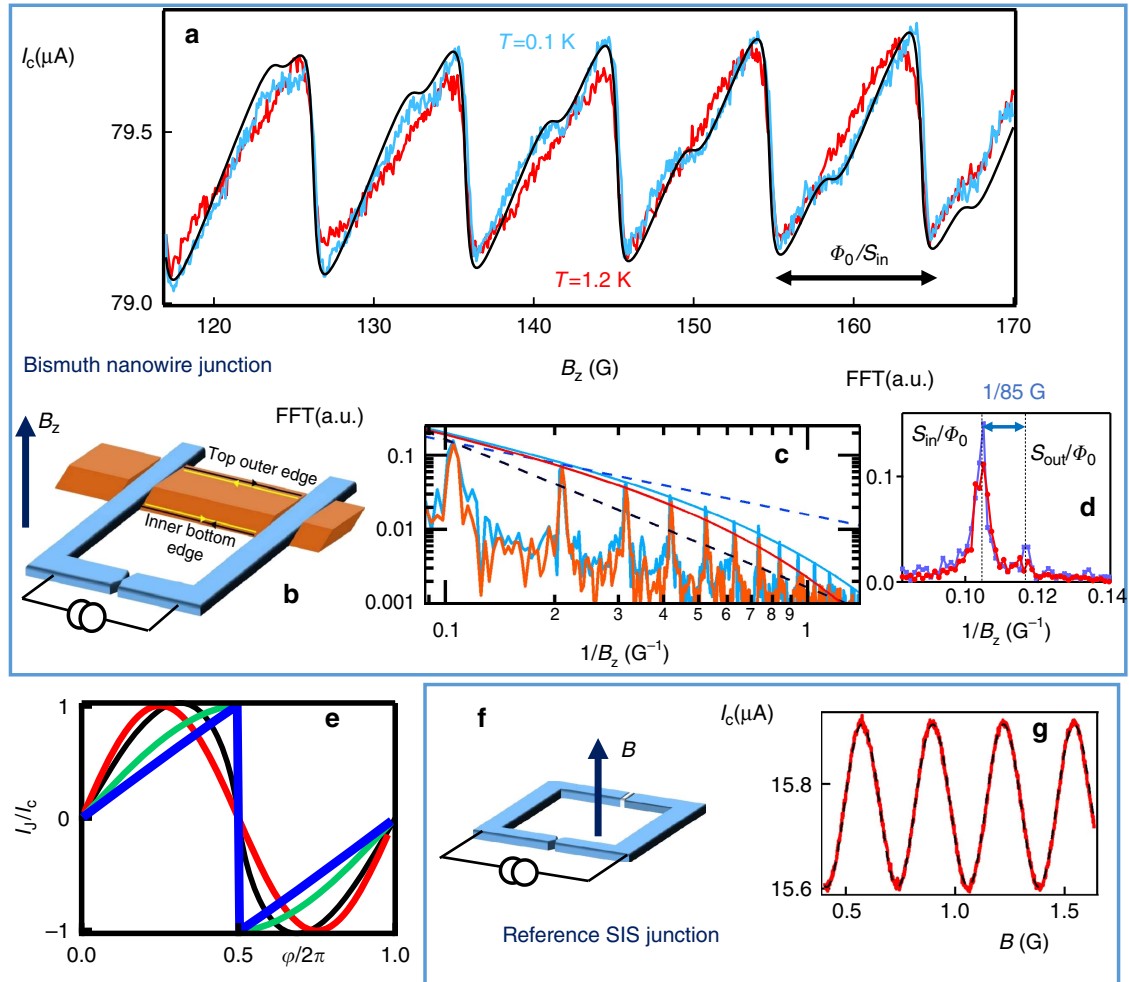

**Figure 2 | Sawtooth-shaped CPR of the S/Bi/S junction.** (**a**) Critical current of the asymmetric SQUID, at 130 mK (blue) and 1 K (red), revealing the bismuth junction's CPR and combination of two sawtooth-shaped currents $\sum(-1)^n/n\{\sin(n\varphi)\exp(-0.19n)+0.25\sin(1.1\times n\varphi)\exp(-0.45n)\}$ (black). Here $\varphi=2\pi B_z S_{in}/\Phi_0$, with $\Phi_0=h/2e$ and $S_{in}$ is the area delimited by the SQUID loop and the inner edge of the wire. (**b**) Sketch of the asymmetric SQUID made of the tungsten nanoconstriction in parallel with the Bi nanowire, with its two edge states running along the top outer edge and inner bottom edge. (**c**) Fourier transform of the measured curves at 130 mK and 1 K. We have included the $1/n$ dependence expected for the harmonics of the pure sawtooth (dashed blue line) as well as the phenomenological fits to the decay of these harmonics. We also show the $1/n^2$ dependence of a diffusive junction (dashed black line). (**d**) Satellite peak on the CPR's Fourier transform due to the second conduction path at the outer edge of the wire (delimiting an area $S_{out}$), next to the main peak (corresponding to the area $S_{in}$). (**e**) Theoretical zero-temperature CPRs of superconductor/normal metal/superconductor junctions from tight-binding simulations on a square lattice[40] in different regimes: green, short junction with one N site, the CPR is close to the ideal relation $I_s=2e\pi\Delta/h\sin(\varphi/2)\,\mathrm{sign}(\pi{-}\varphi)$; blue, sawtooth CPR of a long single-channel ballistic junction, calculated for length $L=1.2\times\xi_s$. Black: CPR of a multichannel diffusive wire $\sum(-1)^n/(2n+1)(2n-1)\sin(n\varphi)$[32]. Red, sinusoidal CPR $I_s=e\pi\Delta/h\sin\varphi$ of a multichannel SIS junction with normal state resistance $h/2e^2$. All curves are normalized to the maximum value of the CPR. (**f**) Sketch of the reference asymmetric SQUID, made of a similar tungsten constriction in parallel with a superconducting aluminium/oxide/aluminium SIS tunnel junction. (**g**) Critical current of the reference asymmetric SQUID, yielding, as expected, a sinusoidal CPR (dashed line is a sinusoidal fit to the data). The small effect of the circuit inductance has been corrected for. This correction is negligible for the Bi-SQUID because of its smaller dimensions.

of a long, perfectly connected ballistic channel. This is a key result of our paper. Eleven harmonics are visible in the Fourier transform of the signal at 100 mK. Figure 2b displays how the $1/n$ amplitude of the $n$th harmonic, $I_n$, expected for a perfect sawtooth, is exponentially damped by a small factor: $I_n=I_0/n\exp(-an)$, with $a=0.19$ at 100 mK and 0.25 at 1.2 K. Since the $n$th harmonics can be interpreted as the contribution to the supercurrent of the $n$th order Andreev reflection, the damping can be understood as due to an imperfect transmission encountered $n$ times, as well as due to temperature $T$ (refs 11–13), yielding the phenomenological expression $I_n=I_0/nt^{2n}\exp(-nT/E_{Th})$. Here $t$ is the modulus of the total transmission in the normal state. Other effects, such as high

frequency noise, may also contribute to this decay. The damping coefficient at 100 mK yields a quasi perfect transmission $t\gtrsim0.9$, and we deduce the Thouless energy $E_{Th}=4\,K$ from the damping coefficient at 1.2 K.

**Signatures of a second supercurrent-carrying path.** Predicted by our tight-binding modelling of the wire (Fig. 1c), a second path can be identified upon closer inspection of the CPR: a wiggle with a smaller period is superimposed to the main sawtooth signal. The full critical current modulation is reproduced by adding to the main sawtooth the contribution of a second sawtooth with a four times smaller amplitude and a 10% smaller period than the

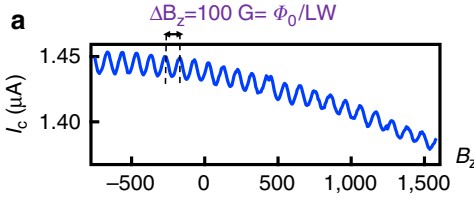

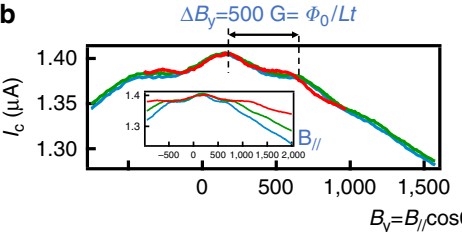

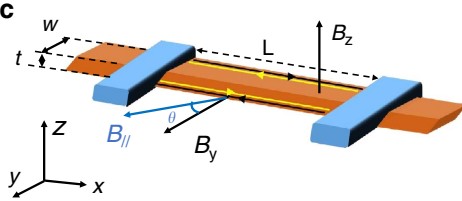

**Figure 3 | Interference between edge states in simple two-wire geometry.** The oscillations of the critical current of the sample connected in a two-wire configuration, before integration in the asymmetric SQUID configuration, confirm that the supercurrent-carrying paths run along diagonally opposite sample edges. Both in-plane and out-of-plane magnetic fields generate oscillations of the critical current, with periods of 100 G for an out-of-plane field $B_z$ (**a**), and 500 G for an in-plane field $B_y$ perpendicular to the wire axis (**b**). The factor of 5 between the two periods roughly coincides with the ratio of the wire width over wire thickness, confirming that the two interfering paths are along diagonally opposite edges of the wire. **b** and inset: different curves in an in-plane field $B_{//}$ at different angles overlap when the field is renormalized by the cosine of its angle with respect to the $y$ axis. We note that the critical current of the Bi nanowire is about two times larger than the current measured in the asymmetric SQUID set-up. This change can be attributed to thermal cycling of the sample. It may also be a signature of a topological crossing at $\varphi = \pi$, that would lead to a two times greater critical current measured in a current-biased set-up compared to the critical current measured in a phase-biased set-up[13]. (**c**) Sketch of the wire and orientations of the magnetic field.

first. This indicates that this second, ballistic, path is situated along the outer wire edge. The sum of these two sawtooths generates a beat period of $\sim 85$ G (Supplementary Fig. 7 and Supplementary Note 4) which corresponds to a flux quantum through the wire. This second path is also apparent as a satellite peak in the Fourier transform next to the first harmonic (Fig. 2d), whose exponential decay $\exp(-0.45n)/n$ yields a transmission coefficient $t \sim 0.75$. The smaller supercurrent carried by the second channel could be due to a disordered region between the contact and this edge (inset Fig. 1a).

These findings of two distinct current paths, located at two opposite edges of the wire, are consistent with the two-path interference patterns of the critical current measured on the same wire before integration in the asymmetric SQUID configuration, for different orientations of the magnetic field (Fig. 3). The oscillation periods depend on the field direction, and correspond to one flux quantum through the surface enclosed by the two paths projected onto a plane normal to the field. It is

thus consistent that the 100 G interference period in a perpendicular field $B_z$ (corresponding to $\Phi_0/WL$) is the same (given the uncertainty of the sample position in the magnet) as the beat period measured in the CPR (Supplementary Fig. 7). The critical current oscillations in an in-plane magnetic field $B_{//}$ was also measured for several angles. The period is determined only by the projection of the field on the normal to the wire axis, that is, the $y$ component. The period $\Delta B_y = 500$ G $= \Phi_0/tL$ yields a nanowire thickness of $t = 40$ nm, consistent with scanning electron microscope observations. We note that oscillating critical current patterns were also detected previously in Bi wires with unknown crystalline orientations[25], and were also interpreted as interference between two paths through the samples[26].

**Number of channels and Josephson current amplitude.** The amplitude of the measured supercurrent as well as its resilience up to high magnetic fields ($B_{max} = 0.5$ T) (Fig. 3, Supplementary Fig. 6 and ref. 25) are consistent with a small number of channels, each of them confined to an extremely narrow region in space (within $< 4$ nm $= \Phi_0/B_{max}L$). The maximum supercurrent through one ballistic channel is $I_1 = \pi\Delta/\Phi_0 = e\Delta/2\hbar \sim 250$ nA for a short junction (that is, much shorter than $\xi_s = \hbar v_F/\Delta \approx 600$ nm). It is smaller for a long junction[13], of the order of $ev_F/L \approx 100 \pm 30$ nA for $L = 1$ µm and $v_F = 6 \pm 2 \times 10^5$ m s$^{-1}$. This value of $v_F$, deduced from the Fourier spectrum of the CPR at 1 K, is in qualitative agreement with the values deduced from the dispersion relation of 1D edge states of (111) Bi triangular surfaces, measured by photoemission in ref. 27. The critical current of the nanowire, given by the modulated current amplitude of 400 nA (Fig. 2a), thus implies that at most six perfectly transmitted channels carry the supercurrent. A reasonable assumption is that one path contains three to four quasi perfect channels, each with the same sawtooth-shaped CPR, and all situated at the inner edge of the wire, on the bottom (111) facet. They could be associated to the orbitals $p_x, p_y, p_z$ of Bi, as suggested by Murakami[17], or could also run along the edges of few parallel terraces at the facet edge. The smaller contribution of the second path is attributed to one or two other channels of smaller transmission, at the outer edge of the top (111) facet (see sketch in Fig. 2b).

**Phase shifts of the CPR induced by an in-plane magnetic field.** The purely 1D nature of the edge Andreev bound states that carry the supercurrent across the nanowire implies that they are insensitive to orbital dephasing and offers the possibility to explore the effect of a Zeeman field on the phase of the Josephson current. A Zeeman field can induce a crossing of the Andreev levels, turning an energy maximum into a minimum[28]: this causes a sign change of the CPR, or equivalently a $\pi$ shift of its phase. $0-\pi$ transitions are expected when dephasing by the magnetic field equals dephasing by the propagation time through the wire, that is, when the Zeeman energy equals the Thouless energy, $g_{eff} \mu_B B = \hbar < v_F > /L$. Such $0-\pi$ transitions are visible in Fig. 4a,b,e, as phase jumps in the CPR plotted as a function of a magnetic field in the (111) plane, either perpendicular or parallel to the wire axis. The characteristic field $B_{x,y} \sim 600$ G, $B_z \sim 400$ G between two successive $0-\pi$ transitions yields an effective $g$ factor $g_{eff} \approx 30$–100, consistent with the high $g$ factors of some bands in Bi (ref. 29) as well as recently found for the surface states with ARPES experiments[30] in high magnetic field. We note that penetration of vortices in the superconducting electrodes would also lead to phase jumps. This is, however, unlikely as no sign of hysteresis was found in our data. This realization of a $0-\pi$ transition induced by the Zeeman field is possible because the junction is long, contains few channels and the $g_{eff}$ are high

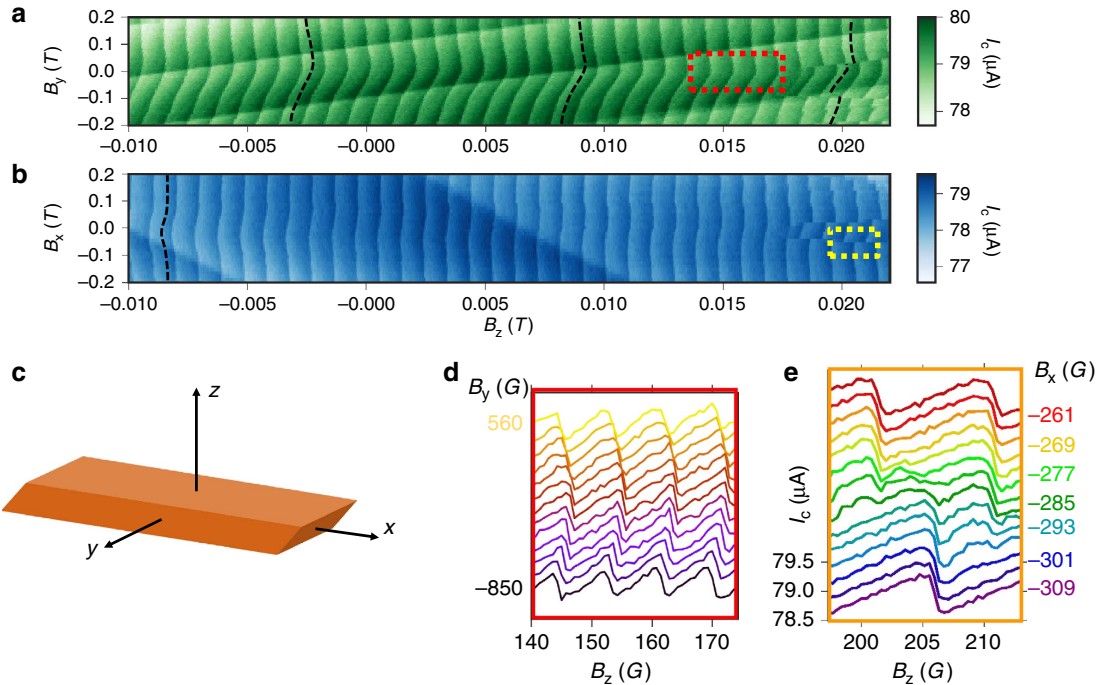

**Figure 4 | Zeeman-induced $\varphi_0$-junctions and 0–π transitions. (a,b)** Colour-coded critical current of the asymmetric SQUID at 100 mK, as a function of both a magnetic field $B_z$ perpendicular to the (111) surface, and in the (111) surface plane, either perpendicular (**a**) or parallel (**b**) to the wire axis. The critical current variations are exactly the CPR of the bismuth nanowire. The dashed lines outline how the phase of the CPR shifts with magnetic field ($\varphi_0$ junctions), and in some instances, jumps by π (0–π transitions). Note that a linear relation between $B_x$ ($B_y$) and $B_z$ was subtracted, to account for the small (few percents) projection of the $B_z$ field on the horizontal sample plane. This treatment eliminates the linear variations of the CPR's phase with magnetic field. (**c**) Sketch of the wire orientation. (**d**) Sections of the colour-coded plot displaying how the $B_y$ field shifts the phase of the CPR, generating $\varphi_0$ junctions. (**e**) Sections of the colour-coded plot as the $B_x$ field causes a 0–π transition.

enough that the transition occurs at a magnetic field below the superconducting electrodes' (relatively high) critical field.

The usual conversion of phase difference into a charge (super-) current can be supplemented by a conversion into a spin current in the presence of strong spin–orbit interaction. Consequently, a magnetic field can shift the CPR by a phase $\varphi_0$ (refs 15,26). This effect is related to the spin-splitting of Andreev states induced by spin–orbit interactions, that generates spin-dependent velocities, and is expected to be enhanced in quantum spin Hall ribbons with asymmetric edges[16]. It is also predicted to be greatest in a field parallel to the spin–orbit field. Such '$\varphi_0$-junctions' are visible in Fig. 4a,b,d, in the form of an oscillating phase of the CPR with increasing field, outlined by the winding hatched curve, with a stronger effect in a field perpendicular to the wire axis (compare Fig. 4a,b).

## Discussion

The CPR of a Bi-nanowire-based Josephson junction demonstrates that conduction occurs along ballistic channels confined at two edges of the wire's (111) facets. Whereas narrow edge states were predicted for a bilayer of (111) Bi (ref. 17), it was not obvious that such states should exist in thicker samples, until 1D edge states were detected by scanning tunnelling microscopy recently, in two layer-deep pits at the (111) surface of bulk Bi crystals[31]. Our work shows that in the superconducting proximity regime, the contribution of the nanowire's two ballistic edge states outweighs that of the more numerous diffusive channels on the wire's non-topological surfaces. That is because the contribution to the supercurrent is proportional to the conductance times Thouless energy[32], leading to a critical current of $I_{c,1ch,diff} \propto l_e^2/L^3$ for a diffusive channel (with $l_e$ the mean free path), and

$I_{c,1ch,bal} \propto 1/L$ for a ballistic channel. Thus a diffusive channel carries a supercurrent smaller than a ballistic channel by a factor $(L/l_e)^2 \sim 100$, where $l_e$ is the mean free path. Moreover, in addition to being ballistic rather than diffusive, topological edge states are better coupled to the superconducting electrodes due to their spin-moment locking that should produce perfect Andreev retro-reflections[33]. By contrast, the slower diffusive carriers also undergo regular (as opposed to Andreev) reflections on opaque barriers, which decrease the supercurrent they can carry. Our results show that proximity-induced superconductivity can be exploited to reveal and select ballistic protected edge states in systems where topological and non-topological conduction paths coexist. Moreover, even if the topological character of the edge states carrying the supercurrent is not yet directly proven, our results indicate an effective mean free path for the edge states much greater than 1 μm, whereas the mean free path of the surface and bulk states, whose contribution dominates the transport in the normal state, does not exceed 200 nm (Supplementary Note 1). This result in itself points towards a topological protection for these edge states. We are now working on microwave experiments on these same wires that, by exploring the nature of the level crossing at $\varphi = \pi$, should be more definite on this issue[34–36].

## Methods

**Sample fabrication and measurement.** Bismuth nanowires were grown by RF-sputtering a Bi target of 99.9999% purity onto Si substrates at 473 K and in an argon pressure of 10 mBars. A Bi film of 400 nm deposited at a rate of 0.7 nm s⁻¹ in these conditions exhibits sparse arrays of nanowires, typically 100 μm long and distant by 10 μm from one another. High resolution transmission electron microscope observations indicated high quality single crystals, of rhombohedric or hexagonal sections as well as clear facets, with typical width of 50–300 nm. The nanowires were then dry deposited on an oxidized silicon substrate. After optical

selection, their crystalline orientation was determined using electron backscatter diffraction. An example is shown in Supplementary Fig. 2. This characterization was performed at the extremity of the nanowires to avoid electron beam-induced damage of the Bi wire. We also checked on similar nanowires that their crystalline orientation was constant as we changed the position of the beam spot, as is expected for single crystals. At this stage, we selected nanowires whose top surface was determined to be oriented perpendicular to the trigonal [111] axis. Electrical connections were then made using gallium focused ion beam (FIB)-deposited superconducting tungsten wires[37]. For all the measured nanowires, the lengths between the $W$ lines were chosen to be greater than 1 µm to avoid possible superconducting contamination. Previous studies using the same set-up showed that this could be an issue for wires whose length is below 200 nm. For the same reasons, we minimized the total exposure time of the nanowires under the FIB to a single scan at high scanning rate and low magnification. We connected nine segments with different lengths, from a total of three such nanowires. Their resistances and lengths are summarized in the Supplementary Table 1, and show low contact resistance on average.

The samples were then cooled to 100 mK and their critical current versus magnetic field was measured using a lock-in detection technique.

After this first characterization step, the CPR was measured on one segment ($s_1$JU segment of Supplementary Table 1) using the asymmetric SQUID technique[20] with a reference junction made of a W constriction. To this end, a FIB-deposited W wire was added between the two tungsten electrodes, in parallel to the Bi nanowire (Fig. 1), and was subsequently etched with the Ga + beam while measuring the total resistance between the contacts, until the total resistance reached 190 Ω, corresponding to a constriction resistance of 300 Ω. The critical current of the SQUID was deduced from the average of 100 to 400 measurements of the switching current, using a counter synchronized to a current ramp at 180 Hz, and triggered by the jump in the sample resistance. In those measurements, the sample resistance was measured at 100 kHz with a lock-in detector operating with a time constant of 1 ms.

**Tight-binding simulation of bismuth nanowire.** Tight-binding simulations were performed on a 1D infinite geometry corresponding to the stacking of five (111) Bi bilayers, seven atoms wide, arranged to provide a simplified model of the rhombic section of the nanowire studied in this paper. The z direction therefore corresponds to the [111] direction or trigonal axis in Bi's rhombohedral lattice, which was determined experimentally by electron backscatter diffraction. The infinite direction of the nanowire and the direction of the lateral surface were chosen in order to match two criteria: they should be of high symmetry since nanowire growth mechanisms favour high symmetry surface orientation, and should reproduce the rhombic section that was inferred from scanning electron microscope observation. The retained geometry corresponds to the (100) lateral surface and a growth direction along the bisectrix axis. This corresponds to the case studied by Murakami in the 1 bilayer thickness limit and 20 atoms wide. We find a sharp density of states along two edges, that correspond to zigzag type A edges (defined in ref. 31) on the top surface, and B type edges on the bottom surface: those are the edges whose atoms are less coupled to the bulk.

The Hamiltonian matrix was written according to the $sp3$ tight-binding model of Liu and Allen[38] using the KWANT python package[39]. The local density of states at the Fermi level was computed by summing the local spectral weights in a window of 0.1 eV around the Fermi level. A smaller window would yield even higher densities of states on the acute edges.

**Data availability.** All data presented in the main paper and supplement are available from the authors upon request.

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

## Acknowledgements

We acknowledge fruitful discussions with M. Aprili, S. Bayliss, C. Beenakker, S. Bergeret, D. Carpentier, J. Jobo, M. Houzet, F. Konschelle, J. Meyer, J. Enrique Ortega, P. Simon, T. Wakamura, A. Yazdani and financial support from CNRS, ANR MASH (ANR-12-BS04-0016), ANR DYMESIS (ANR-2011-IS04-001-01), ANR DIRACFORMAG (ANR-14-CE32-0003) and ANR JETS (ANR-16-CE30-0029-01). Yu.A.K. acknowledges financial support by RFBR and Moscow region grant 14-48-03664.

## Author contributions

A.M., R.Del., R.Deb., H.B. and S.G. conceived the experiment and performed the measurements. A.K., Yu.A.K., V.T.V. and I.I.K. grew the Bi nanowires and characterized the nanowires with transmission electron microscope, A.M., A.K., S.S. and S.G. fabricated the circuits using FIB, F.B. and A.M. characterized the crystalline orientation of the nanowires. A.M., H.B. and S.G. wrote the manuscript with input from all authors. A.C., A.M. and H.B. conducted the numerical simulations.

## Additional information

**Competing interests:** The authors declare no competing financial interests.

**Publisher's note**: 

