## [Peer Review File · Nature Communications]

Reviewers' comments:

Reviewer #1 (Remarks to the Author):

In the manuscript authors have studied a system of Bismuth nanowires in a SQUID and showed experimental evidence for ballistic quasiparticles in the observed CPR. The key experimental observation is the linear sawtooth CPR, characteristic of ballistic SNS junction, doesn't appear to have been ever observed on any 1D system exceeding micrometer in length.

The central claim of the paper has been checked against current literature, and, indeed, it appears that in none of the experimentally available systems including proximitized III-V semiconductors (InAs/GaSb, 2DEGs, InAs, core-shell Al/InAs, and InSb nanowires), HgTe/CdTe QW edge states, edge states of graphene, and 1D states in carbon nanotubes, has the linear CPR been measured. As such, the references given in the second paragraph of the manuscript appear to be comprehensive.

Claims of ballistic edge states have been, however, made in <http://dx.doi.org/10.1038/ncomms8426>, but the size of the system is small, and no linear CPR has been measured.

A somewhat linear CPR has been, however, observed on a 500um long CNTs in <http://www.nature.com/nnano/journal/v1/n1/pdf/nnano.2006.54.pdf> (Figure 5a)

A weaker statement in the manuscript is the count of the states that are responsible for the observed behavior - "three to six perfectly transmitted channels carry the supercurrent...one path contains three to four quasi-perfect channels". Is it possible to understand that number based on the number of left- and right- movers in the band structure of the wire and Fermi velocities? (also see question 2 below)

A strong experimental point is made about the supercurrent paths localized to the edges, as the CPRs have two distinct periodicities corresponding to two current paths with different 10% area of the SQUID loop.

Overall, I find this work quite impressive, particularly with the number of supplementary experiments performed (temperature dependence, field direction dependence, critical current measurements, multiple wires, and comparison to normal-metal SNS junctions). I recommend the manuscript for publication in Nature Communications, however, I would like to pose the following questions that, I hope, would improve the manuscript:

Question 1:

line 75,76 & Fig 1A inset - Please explain clearer why the outer edge is more disordered/coupled worse to the superconductor. Has this behavior systematically been observed on multiple wires? Is it possible that the outer edge simply gets damaged more during the fab than the inner one? Is it possible to have a different device, in which the inner and outer edge are flipped relative to the SQUID loop?

line 222 - "the SEM image indeed suggests that the W electrode may not contact the bottom nanowire facet". Doesn't that statement contradict that the higher supercurrent is carried by the inner nanowire facet?

Question (2):

Fig 2C - "theoretical zero-temperature CPR of SNS junctions from tight binding". Does this mean tight-binding model of Bi nanowire, or some other generic SNS junction? In other words, has a simulation for the disordered Bi nanowire SNS junction been carried out with a realistic band

structure of Bi (described in section "Tight binding simulation", line 228)? If so, please add this to the Supplementary S7.

Question (3):

Line 104. Does the effective g-factor of 30-100 correspond to g-factor of 1D edge states? For those highly localized states, one would naively expect g-factors closer to the vacuum value of 2. The g-factor of about 30 has recently been measured on Bi(111), and is smaller than that of the bulk due to dimensional reduction:

<http://www.nature.com/articles/ncomms10814>

In 1D, one would thus expect an even lower number. Please comment on that.

Question (4):

Has an AFM study of the wires/devices before and after fab been carried out? Is it possible to assess the quality of wire facet's surfaces, surface oxidization, and interfaces with superconductor using an atomically resolved tool?

Question (5):

Fig 2B, inset. Can a similar graph be plotted for 1.2K? The signal from outer edge states appears to be more destroyed by this small temperature. A comment on that observation has to be made in the main text of the manuscript, as it appears to be related to the poor visibility of the CPR from the outer edge.

Question (6):

Fig 3A and B, shows additional periodicity in B_y and B_x respectively - about 0.2T in B_y . Can that periodicity be explained by the geometry of the nanowire and its current distribution? E.g., does that B_y periodicity measure the flux through XZ plane? Does B_x periodicity correspond to some sort of misalignment?

Small remarks (probably not exhaustive list):

Line 41, why the (100) states are called "diffusive"?

Line 52 "L is the N length ..." seems colloquial, change to "where L is the length of the normal-metal section, v_f is the Fermi velocity, and Δ is the superconducting gap"

Line 65, 74, 83, etc. "Fig. XX", capital

Line 156, 171 SQUID, capital

line 171 both -a- magnetic field B_z

Figure S5 - Reference is written in French.

Please proof-read the manuscript and supplementary.

Reviewer #2 (Remarks to the Author):

The manuscript of Anil Murani et. al. reports on SQUID measurements of conducting Bismuth nanowires. They provide a direct signature of ballistic 1D transport along the topological surfaces of a single crystal bismuth nanowire connected to superconducting electrodes. To do so, they exploit the extreme sensitivity of the relation between the Josephson current I_J flowing through a nanostructure and the superconducting phase difference at its ends, the "current-phase relation". As pointed out by the authors, the sharp sawtooth-shaped CPR demonstrates that transport occurs ballistically along two edges of the nanowire, and confirms the predicted nearly perfect transmission of Cooper pairs through Quantum Spin Hall edge states

In my opinion, the originality and broad interest of this study is fully established and warrant its publication in "Nature Communications". However, I would like to question below the data and

methodology of the present study, which may strongly impact the conclusions of the article. I hope the following remarks will be useful for the authors in order to improve their manuscript.

1/ As displayed in Fig. 2A, the oscillatory I_c curves at $T=0.1K$ have two different periods: "A wiggle with a smaller period is superimposed to the main sawtooth signal." The author's numerical simulation predicted that "this second, ballistic, path is situated along the outer wire edge, since the area delimited by the two wires edges is one tenth of the SQUID loop area." As I know, this secondary period can always be caused by the quality of the superconducting electrodes. The author should be very careful to make this conclusion, since the author has no extra data that could support this point of view. To confirm this, a control experiment is needed, such as changing the loop area S_{ext}/S_{int} .

2/ The author says that "The wires are narrow enough (thickness and width between 100 and 200 nm) that most normal state conduction is due to surfaces and edges, with only a small contribution of the bulk states." In my opinion, the diameter 100~200 nm is not small enough that to eliminate the influence of the contribution of the bulk conduction. The author should provide more data with different values of the nanowire diameter, instead of merely varying the sample length, to clarify this point.

As a general conclusion, I find the manuscript of Anil Murani et. al. very interesting and novel. It paves the way towards Superconductor/topological insulator/Superconductor devices and should inspire many future works relating the study of Cooper pairs transmit through QSH edge states. However, in my opinion the authors should be more careful with respect to some of their conclusions, and control-experiments are needed in order to avoid making any incorrect conclusions.

Reviewer #3 (Remarks to the Author):

The manuscript describes experiments with Josephson junction made of a single crystal Bi nanowire and W superconductors deposited with focused-ion-beam. Current phase relationship (CPR) of the junction was measured by embedding it in an asymmetric SQUID by attaching W nano-constriction in parallel as a reference Josephson junction. One of the main results of this paper is the observation of two saw-tooth shaped CPR with slightly different periodicity. Authors claim that there are two ballistic and topological 1-dimensional conduction states which are positioned at inner and outer edges of the Bi nanowire, and these states mediate long ballistic Josephson coupling between W superconductors. In addition, by exploiting strong spin-orbit coupling in Bi nanowire, 0- π transition and ϕ_0 -junction behaviour were demonstrated by applying magnetic field.

The data is well analysed and the paper is well written. However, I have several questions which the authors should address before I can recommend publication.

1. Saw-tooth shaped CPR which was predicted for long-ballistic Josephson junction implies that the edge conducting channels are ballistic. However, the topological properties of these channels are not much discussed in the paper. Although it's theoretically predicted that topological conducting channels produce perfect Andreev reflections, nearly perfect transmission coefficient that was experimentally measured in the paper does not necessarily means the conducting channels are topological. What is the experimental evidence that these edge conduction is topologically protected?

2. Authors claim that Bi nanowire shows quantum spin Hall edge states in line #28. If it's QSH, shouldn't conductance be $2e^2/h$? Can authors discuss how many edge conducting channels are theoretically expected in such Bi nanowire system?

3. At line #75, authors say that the supercurrent carried by outer edge state is smaller probably because outer edge is more disordered. It is not clear why outer edge state would be more disordered. It is not clear what is the meaning of white arrow pointing inner edge in the inset of Fig. 1A.
4. At line #63 and Fig. 2B, it is hard to see the exponential dependence of $\exp(-0.19n)$ for $T=0.1\text{K}$ and $\exp(-0.45n)$ for $T=1.2\text{ K}$ in Fig. 2B. Can guide lines be drawn in Fig. 2B to show clear exponential decay behaviour?
5. From line #65 and Supplementary section S6, authors claim that the transmission coefficient is close to 1 according to the analysis of exponential decay of FFT amplitude of CPR. Authors need to explain how this analysis works. Why amplitude of n -th harmonics decreases as $t^{\{2n\}}$ and as $\exp(-n(T/E_{\text{Th}}))$ at higher temperature? Is there any reference related to this analysis?
6. Reference 11 is not correctly cited in line #41 and line #87. Reference 11 does not have 'Murakami' as an author.
7. Methods section describing how to produce nanowires is not clear enough for other researchers to reproduce. It is needed for authors to describe more details or add related references?
8. The word 'extreme' in line #24 should be removed, or authors should describe how sensitive CPR is more quantitatively or qualitatively.

We thank the three referees for their careful reading of our manuscript, and their remarks and questions, that have been helpful for us to improve the manuscript. We have divided the “results” section using subheadings, in accordance with the format of Nature Communications, which we think improves the clarity of the paper. We answer their questions below and indicate the changes we have made to the manuscript.

Reviewer #1 (Remarks to the Author):

In the manuscript authors have studied a system of Bismuth nanowires in a SQUID and showed experimental evidence for ballistic quasiparticles in the observed CPR. The key experimental observation is the linear sawtooth CPR, characteristic of ballistic SNS junction, doesn't appear to have been ever observed on any 1D system exceeding micrometer in length. The central claim of the paper has been checked against current literature, and, indeed, it appears that in none of the experimentally available systems including proximitized III-V semiconductors (InAs/GaSb, 2DEGs, InAs, core-shell Al/InAs, and InSb nanowires), HgTe/CdTe QW edge states, edge states of graphene, and 1D states in carbon nanotubes, has the linear CPR been measured. As such, the references given in the second paragraph of the manuscript appear to be comprehensive. Claims of ballistic edge states have been, however, made in <http://dx.doi.org/10.1038/ncomms8426>, but the size of the system is small, and no linear CPR has been measured. A somewhat linear CPR has been, however, observed on a 500um long CNTs in <http://www.nature.com/nnano/journal/v1/n1/pdf/nnano.2006.54.pdf> (Figure 5a)

We agree that non sinusoidal current phase relations in carbon nanotubes have also been measured. However the physics is quite different. In the work mentioned by the referee, the non-harmonic current phase relation is measured in the vicinity of a zero/pi gate-voltage-dependent transition induced by the spin state of the carbon nanotube in the Kondo regime. In that regime, explored more precisely in ref (23) , the current phase relation exhibits a first order transition with a sign reversal at a characteristic phase. The CPR is not really linear but rather exhibits a sharp pi phase shift at zero temperature, at the characteristic phase of the spin-state transition of the nanotube. We have now added the reference suggested by the referee in the paper (ref 22).

A weaker statement in the manuscript is the count of the states that are responsible for the observed behavior - "three to six perfectly transmitted channels carry the supercurrent...one path contains three to four quasi-perfect channels". Is it possible to understand that number based on the number of left- and right- movers in the band structure of the wire and Fermi velocities? (also see question 2 below).

Our tight binding numerical simulations on very small sizes are very useful for the indication of the existence of edge states in the nanowire. It is however difficult to extract from them quantitative information on the Fermi velocity and amplitude of wavefunctions on the edges. More extensive ab initio calculations are clearly needed, but they are beyond the scope of this experimental paper.

However considering the existence of 3 delocalised orbitals px, py and pz per Bi atom, it is reasonable to assume, following Murakami, that 3 conducting channels participate to an edge state.

A strong experimental point is made about the supercurrent paths localized to the edges, as the CPRs

have two distinct periodicities corresponding to two current paths with different 10% area of the SQUID loop.

Overall, I find this work quite impressive, particularly with the number of supplementary experiments performed (temperature dependence, field direction dependence, critical current measurements, multiple wires, and comparison to normal-metal SNS junctions). I recommend the manuscript for publication in *Nature Communications*, however, I would like to pose the following questions that, I hope, would improve the manuscript:

Question 1: line 75,76 & Fig 1A inset - Please explain clearer why the outer edge is more disordered/coupled worse to the superconductor. Has this behavior systematically been observed on multiple wires? Is it possible that the outer edge simply gets damaged more during the fab than the inner one? Is it possible to have a different device, in which the inner and outer edge are flipped relative to the SQUID loop? zoom of the contact area, explaining the asymmetry between the two conduction channels.

line 222 - "the SEM image indeed suggests that the W electrode may not contact the bottom nanowire facet". Doesn't that statement contradict that the higher supercurrent is carried by the inner nanowire facet?

The upper (111) surface of the nanowire, which contains the outer edge, has been etched and therefore likely damaged by the FIB. The W electrode penetrates deeply into the bismuth nanowire, but probably does not reach the bottom surface, as noticed by the referee. This is why we believe that the bottom, inner edge is not damaged but yet well coupled to the W wire. This is now specified in the caption of Fig. 1, where we write "The (111) surface has been etched by the FIB whereas the bottom surface is probably not damaged and thus better coupled to the W superconducting wire."

Up to now we have only measured two SQUIDS and observed clearly the period beating between the 2 periods corresponding to the 2 edges on only one of them. We have however observed a modulation with a period corresponding to the area of the nanowire on several other samples in a simple SNS geometry as discussed in S4 and reference Li et al. PRB(2014). Those results support the fact that supercurrent is carried by edge states at opposite sides of those wires as well.

Question (2): Fig 2C - "theoretical zero-temperature CPR of SNS junctions from tight binding". Does this mean tight-binding model of Bi nanowire, or some other generic SNS junction? In other words, has a simulation for the disordered Bi nanowire SNS junction been carried out with a realistic band structure of Bi (described in section "Tight binding simulation", line 228)? If so, please add this to the Supplementary S7.

A full model of a tight binding calculation of the Bi nanowire in a SNS configuration is beyond the scope of this paper. The numerical simulations of the CPR we show are performed on model systems with a square lattice in a tight-binding approximation (Fig 2C), or hexagonal lattice with Kane and Mele model in S7. We use only one orbital per site, which is not sufficient to fully describe Bi. We now specify the model we use in the caption of Fig 2C and S7.

Question (3): Line 104. Does the effective g-factor of 30-100 correspond to g-factor of 1D edge states? For those highly localized states, one would naively expect g-factors closer to the vacuum value of 2. The g-factor of about 30 has recently been measured on Bi(111), and is smaller than that of the bulk due to dimensional reduction: <http://www.nature.com/articles/ncomms10814> In 1D, one would thus expect an even lower number. Please comment on that.

We thank the referee for bringing this reference to our attention, since there is very few available data on the g factor for surface states. We have added this reference to the paper. We were also surprised by the large g values needed to explain our data. However we did not find any predictions on the g factor values expected for the edge states on Bi(111) surfaces.

Question (4): Has an AFM study of the wires/devices before and after fab been carried out? Is it possible to assess the quality of wire facet's surfaces, surface oxidization, and interfaces with superconductor using an atomically resolved tool?

The thickness of the oxidation (few nm) was estimated from the transmission electron microscopy experiments shown in Fig. S1. We have tried to conduct also scanning tunneling microscopy experiments on these nanowires. But we did not succeed due to the bad adhesion of the nanowires to the surface of the metalized wafer on which they were deposited for that experiment. The thickness of the nanowires was estimated from SEM observations. AFM measurements of similar wires were performed, and found consistent with the SEM images. We refrain from performing AFM measurements on the very wires inserted in the squid configuration until the microwave measurements (mentioned at the end of the paper, aiming to detect the topological protection, and quite difficult and requiring a long time to complete) are completed, for AFM measurements can sometimes be destructive.

Question (5): Fig 2B, inset. Can a similar graph be plotted for 1.2K? The signal from outer edge states appears to be more destroyed by this small temperature. A comment on that observation has to be made in the main text of the manuscript, as it appears to be related to the poor visibility of the CPR from the outer edge.

The referee is right, the contribution of the outer edge is not visible at 1.2 K. We have added the data at this temperature on the inset of Fig 2B.

Question (6): Fig 3A and B, shows additional periodicity in B_y and B_x respectively - about 0.2T in B_y . Can that periodicity be explained by the geometry of the nanowire and its current distribution? E.g., does that B_y periodicity measure the flux through XZ plane? Does B_x periodicity correspond to some sort of misalignment?

There is indeed a low frequency field periodicity on the critical current which does not seem to be related to the SQUID periodicity and depends on the orientation of the magnetic field. It was not observed on the Bi wire before insertion in the SQUID. This is why we think that this periodicity does not have anything to do with the Bi nanowire but rather to some defects in the large junction containing the W constriction. It was also not observed in the SQUID shown in Fig S8.

Small remarks (probably not exhaustive list): Line 41, why the (100) states are called "diffusive"?

The referee is right, this adjective is inadequate for numerical simulations done without disorder in the wire. We just had in mind that these states contribute to the experimentally observed diffusive transport in the normal state i.e. with a mean free path, determined in S3, estimated smaller than the length of the nanowire. We now more properly write: "These edge states coexist with 2D metallic states on the non-topological (100) surfaces."

Line 52 "L is the N length ..." seems colloquial, change to "where L is the length of the normal-metal section, v_f is the Fermi velocity, and Δ is the superconducting gap" Line 65, 74, 83, etc. "Fig. XX", capital Line 156, 171 SQUID, capital line 171 both -a- magnetic field B_z , Figure S5 - Reference is written in French.

We have corrected these mistakes.

Please proof-read the manuscript and supplementary.

Reviewer #2 (Remarks to the Author): The manuscript of Anil Murani et. al. reports on SQUID measurements of conducting Bismuth nanowires. They provide a direct signature of ballistic 1D transport along the topological surfaces of a single crystal bismuth nanowire connected to superconducting electrodes. To do so, they exploit the extreme sensitivity of the relation between the Josephson current I_J flowing through a nanostructure and the superconducting phase difference at its ends, the "current-phase relation". As pointed out by the authors, the sharp sawtooth-shaped CPR demonstrates that transport occurs ballistically along two edges of the nanowire, and confirms the predicted nearly perfect transmission of Cooper pairs through Quantum Spin Hall edge states. In my opinion, the originality and broad interest of this study is fully established and warrant its publication in "Nature Communications". However, I would like to question below the data and methodology of the present study, which may strongly impact the conclusions of the article. I hope the following remarks will be useful for the authors in order to improve their manuscript.

1/ As displayed in Fig. 2A, the oscillatory I_c curves at $T=0.1K$ have two different periods: "A wiggle with a smaller period is superimposed to the main sawtooth signal." The author's numerical simulation predicted that "this second, ballistic, path is situated along the outer wire edge, since the area delimited by the two wires edges is one tenth of the SQUID loop area." As I know, this secondary period can always be caused by the quality of the superconducting electrodes. The author should be very careful to make this conclusion, since the author has no extra data that could support this point of view. To confirm this, a control experiment is needed, such as changing the loop area S_{ext}/S_{int} .

We believe we can exclude that the superconducting contact is responsible for the secondary periodic signal, for the following reasons:

First, we have used the same technique to contact silver and gold nanowires and have never observed SQUID-like oscillations. The results on silver are shown in Figure S5.

Second, the control experiment of an asymmetric squid built with a W nanoconstriction and a tunnel junction, with the same tungsten wires forming the squid loop, did not find a secondary periodic signal.

Third, the contribution of two supercurrent-carrying paths running through the entire length of the nanowires was demonstrated in several other wires, of different diameters: Field oscillations of the

critical current of narrower Bi nanowires (below 100 nm diameter) fabricated using a different technique (electrodeposition), were reported in Li et al. PRB (2014). The higher periodicities found (150 and 800 Gauss compared to 100 Gauss in the present experiment) correspond well to two paths carrying the supercurrent at edges of these narrower wires.

In all cases we observe a typically 10% or less modulation amplitude, which indicates that symmetric transmission at the superconducting electrodes is rare. All these periods correspond to rather long trajectories which excludes the contribution of defects localized in the region of the contacts.

Finally, we have added in the main text (new figure 3) an important confirmation, that was initially shown as a supplementary material S4, which validates this spatial distribution of the supercurrent along 2 diagonally opposite edges: The measurement of the oscillation periods of the wire's critical current before insertion in the squid configuration, in different magnetic field directions. The factor of five ratio between the periods in the B_z and B_y directions corresponds to the ratio of the wire between the wire width and thickness, and thus provides confirmation that the interfering paths are located at two diagonally opposite wire edges, that could be the two A-type edges of the nanowire.

2/ The author says that "The wires are narrow enough (thickness and width between 100 and 200 nm) that most normal state conduction is due to surfaces and edges, with only a small contribution of the bulk states." In my opinion, the diameter 100~200 nm is not small enough that to eliminate the influence of the contribution of the bulk conduction. The author should provide more data with different values of the nanowire diameter, instead of merely varying the sample length, to clarify this point.

We completely agree with the referee that it would be very nice to compare Bi samples with different sections. The growing process we use produces nanowires of various rhombohedral or hexagonal sections as seen by SEM and TEM microscopy. However most of the very small section nanowires are lost during the transfer process on to the silicon substrate on which e-beam lithography is done. Note that we have already measured field oscillations in the critical current of different kind of Bi nanowires fabricated with a nearly cylindrical shape using a different technique (electrodeposition) as shown in Li et al. PRB (2014). Periodicities found were 150 and 800 Gauss similar to what is found in the present experiment depending on the field orientation.

The referee is also right that in the 200nm-width wires the contribution of the bulk to normal state transport is not completely negligible. We find in S3 that it contributes to few % of the conductance, and explains the low field magnetoresistance as Shubnikov de Haas quantum oscillations (not shown). We are now more accurate on this point in the manuscript, and write: "most normal state conduction is due to surfaces and edges, with only a few percent contribution of the bulk states (18,19) as discussed in Supplementary materials (20)."

We however could not detect any contribution of these bulk states to the supercurrent. As written in the paper, we expect this contribution to be reduced by $(l_e/L)^2$ per channel compared to the ballistic edge states because of the finite mean free path $l_e = 0.2$ micron (as estimated in S3 from magnetoresistance measurements.)

As a general conclusion, I find the manuscript of Anil Murani et. al. very interesting and novel. It paves the way towards Superconductor/topological insulator/Superconductor devices and should

inspire many future works relating the study of Cooper pairs transmit through QSH edge states. However, in my opinion the authors should be more careful with respect to some of their conclusions, and control-experiments are needed in order to avoid making any incorrect conclusions.

As suggested by the referee comments we have softened our conclusion according to: "Our results show that proximity induced superconductivity can be exploited to reveal and select ballistic protected edge states in systems where topological and non-topological conduction paths coexist. Moreover, even if the topological character of the edge states carrying the supercurrent is not yet directly proven, our results indicate an effective mean free path for the edge states much greater than one micrometer, whereas the mean free path of the surface and bulk states, whose contribution dominates the transport in the normal state, does not exceed 200 nm (20). This result in itself points towards a topological protection for these edge states. We are now working on microwave experiments on these same wires that, by exploring the nature of the level crossing at $\varphi=\pi$, should be more definite on this issue (35,36,37)."

Reviewer #3 (Remarks to the Author):

The manuscript describes experiments with Josephson junction made of a single crystal Bi nanowire and W superconductors deposited with focused-ion-beam. Current phase relationship (CPR) of the junction was measured by embedding it in an asymmetric SQUID by attaching W nano-constriction in parallel as a reference Josephson junction. One of the main results of this paper is the observation of two saw-tooth shaped CPR with slightly different periodicity. Authors claim that there are two ballistic and topological 1-dimensional conduction states which are positioned at inner and outer edges of the Bi nanowire, and these states mediate long ballistic Josephson coupling between W superconductors. In addition, by exploiting strong spin-orbit coupling in Bi nanowire, $0-\pi$ transition and $\varphi 0$ -junction behaviour were demonstrated by applying magnetic field. The data is well analysed and the paper is well written. However, I have several questions which the authors should address before I can recommend publication.

1. Saw-tooth shaped CPR which was predicted for long-ballistic Josephson junction implies that the edge conducting channels are ballistic. However, the topological properties of these channels are not much discussed in the paper. Although it's theoretically predicted that topological conducting channels produce perfect Andreev reflections, nearly perfect transmission coefficient that was experimentally measured in the paper does not necessarily mean the conducting channels are topological. What is the experimental evidence that these edge conduction is topologically protected?

The referee is right saying that we have only proven the 1D and ballistic characters of the states carrying the supercurrent. This is why we are very cautious in the manuscript on this point, avoiding the terminology of topological edge states. Nevertheless we have added the following sentence at the end of the paper: "Our results show that proximity induced superconductivity can be exploited to reveal and select ballistic protected edge states in systems where topological and non-topological conduction paths coexist. Moreover, even if the topological character of the edge states carrying the supercurrent is not yet directly proven, our results indicate an effective mean free path for the edge states much greater than one micrometer, whereas the mean free path of the surface and bulk states,

whose contribution dominates the transport in the normal state, does not exceed 200 nm (20). This result in itself points towards a topological protection for these edge states. We are now working on microwave experiments on these same wires that, by exploring the nature of the level crossing at $\varphi=\pi$, should be more definite on this issue (35,36,37).“

2. Authors claim that Bi nanowire shows quantum spin Hall edge states in line #28. If it's QSH, shouldn't conductance be $2e^2/h$? Can authors discuss how many edge conducting channels are theoretically expected in such Bi nanowire system?

In reference 12 Murakami predicts a conductance quantum of $6e^2/h$ per edge due to the p_x, p_y, p_z orbitals of Bi. The order of magnitude of the critical current we measure is compatible with a conduction of $6e^2/h$ on one edge and $2e^2/h$ on the other. The conductance measured in the normal state is $70e^2/h$, which is much larger. This is due to the contribution of surface and bulk states as discussed in S3. These states however are diffusive and do not give a measurable contribution to the supercurrent in the present experiment.

At line #75, authors say that the supercurrent carried by outer edge state is smaller probably because outer edge is more disordered. It is not clear why outer edge state would be more disordered. It is not clear what is the meaning of white arrow pointing inner edge in the inset of Fig. 1A.

The upper (111) surface of the nanowire which contains the outer edge has been etched by the FIB whereas this is not the case for bottom surface which contains the inner edge. This is why we think that the contact between inner edge is less disordered and for this reason better coupled to the W nanowire which penetrates deeply the Bi wire even if it does not completely connect the edge. This is now specified both in the text and in the figure caption.

4. At line #63 and Fig. 2B, it is hard to see the exponential dependence of $\exp(-0.19n)$ for $T=0.1K$ and $\exp(-0.45n)$ for $T=1.2 K$ in Fig. 2B. Can guide lines be drawn in Fig. 2B to show clear exponential decay behavior?

We have added the exponential behavior of the harmonics in $\exp(-0.19n)/n$ and $\exp(-0.45n)/n$ in Fig.2B.

5. From line #65 and Supplementary section S6, authors claim that the transmission coefficient is close to 1 according to the analysis of exponential decay of FFT amplitude of CPR. Authors need to explain how this analysis works. Why amplitude of n -th harmonics decreases as t^{2n} and as $\exp(-n(T/E_{Th}))$ at higher temperature? Is there any reference related to this analysis? The successive harmonics of the current phase relation correspond to 1D trajectories of length nL where t is the total transmission in the normal state. Their transmissions are therefore expected to decay as t^n at large n , which yields t^{2n} for Andreev pairs. The Thouless energy has been shown to be the characteristic energy scale or the temperature dependence in long junctions by Bardeen et al., that we now refer to in the text in this discussion. We have added the following sentence in the paper to clarify these points and justify the phenomenological expressions we use to fit our data: “ Eleven harmonics are visible in the Fourier transform of the signal at 100 mK. Fig 2B displays how the $1/n$ amplitude of the n th harmonic, I_n , expected for a perfect sawtooth, is exponentially damped by a small factor : $I_n = I_0/n \exp(-an)$, with $a=0.19$ at 100 mK and 0.25 at 1.2 K. Since the n th harmonics can

be interpreted as the contribution to the supercurrent of the n th order Andreev reflection, the damping can be understood as due to an imperfect transmission encountered n times, as well as due to temperature T (11,12,13), yielding the phenomenological expression $I_n = I_0/n t^{2n} \exp(-nT/E_{Th})$. Here t is the modulus of the total transmission in the normal state. Other effects, such as high frequency noise, may also contribute to this decay. The damping coefficient at 100 mK yields a quasiperfect transmission $t \geq 0.9$, and we deduce the Thouless energy $E_{Th} = 4$ K from the damping coefficient at 1.2 K.”

6. Reference 11 is not correctly cited in line #41 and line #87. Reference 11 does not have ‘Murakami’ as an author.

We have amended and corrected the reference list.

7. Methods section describing how to produce nanowires is not clear enough for other researchers to reproduce. It is needed for authors to describe more details or add related references?

We have added a paragraph on the fabrication of the nanowires:

“Bismuth nanowires were grown by RF-sputtering a Bi target of 99.9999% purity onto Si substrates at 473 K and in an argon pressure of 10 mBars. A Bi film of 100nm deposited at rate of 1nm/s in these conditions exhibit sparse arrays of nanowires, typically 100 μ m long and distant by 10 μ m from one another. High resolution TEM observations indicated high quality single crystals, of rhombic or hexagonal sections as well as clear facets, with typical width of 50 to 300 nm.”

8. The word ‘extreme’ in line #24 should be removed, or authors should describe how sensitive CPR is more quantitatively or qualitatively.

We have removed the word “extreme”. The introduction now reads “ To this end, we measure the relation between the Josephson current I_J flowing through a nanostructure and the superconducting phase difference φ at its ends, the “current-phase relation” (CPR) (8). The CPR is an exquisite tool to discriminate between different transport regimes (ballistic, diffusive, tunnel,...). “

REVIEWERS' COMMENTS:

Reviewer #1 (Remarks to the Author):

I believe that authors have addressed both my and the other referee's concerns. Certain issues raised in the referee process - effects of contacts, number of conducting channels, etc. are inherent to the technique employed.

The wording and presentation has been considerably improved. The experimental conclusions are solid and the work presented is interesting enough to warrant publication. I recommend the revised version for publication in Nature Communications.

Reviewer #2 (Remarks to the Author):

Generally, the two questions I raised are well answered by the author, which I think makes the whole story complete and the conclusions solid. Edge conduction is a heated topic, not only because of the novel topological physics, but also the difficulties in signal detection. In this paper, topological Bi nanowires were grown and SQUID measurements were carried out on the CPR aspect. The experimental work is impressive should arouse broad interest in this research area, both experimentally and theoretically. Thus, I recommend this paper to be published in "Nature Communications".

Reviewer #3 (Remarks to the Author):

In the reply as well as the revised manuscript, the authors have clearly answered all my questions and modified the manuscript accordingly. With this, I recommend the publication of the manuscript in Nature Communications.